# Present and Future of Carbapenem-resistant *Enterobacteriaceae* (CRE) Infections

**DOI:** 10.3390/antibiotics8030122

**Published:** 2019-08-19

**Authors:** Beatriz Suay-García, María Teresa Pérez-Gracia

**Affiliations:** Área de Microbiología, Departamento de Farmacia, Instituto de Ciencias Biomédicas, Facultad de Ciencias de la Salud, Universidad Cardenal Herrera-CEU, C/Santiago Ramón y Cajal, 46115 Alfara del Patriarca, Valencia, Spain

**Keywords:** *Enterobacteriaceae*, carbapenem-resistant, CRE, antibiotic resistance, antimicrobials

## Abstract

Carbapenem-resistant *Enterobacteriaceae* (CRE) have become a public health threat worldwide. There are three major mechanisms by which *Enterobacteriaceae* become resistant to carbapenems: enzyme production, efflux pumps and porin mutations. Of these, enzyme production is the main resistance mechanism. There are three main groups of enzymes responsible for most of the carbapenem resistance: KPC (*Klebsiella pneumoniae* carbapenemase) (Ambler class A), MBLs (Metallo-ß-Lactamases) (Ambler class B) and OXA-48-like (Ambler class D). KPC-producing *Enterobacteriaceae* are endemic in the United States, Colombia, Argentina, Greece and Italy. On the other hand, the MBL NDM-1 is the main carbapenemase-producing resistance in India, Pakistan and Sri Lanka, while OXA-48-like enzyme-producers are endemic in Turkey, Malta, the Middle-East and North Africa. All three groups of enzymes are plasmid-mediated, which implies an easier horizontal transfer and, thus, faster spread of carbapenem resistance worldwide. As a result, there is an urgent need to develop new therapeutic guidelines to treat CRE infections. Bearing in mind the different mechanisms by which *Enterobacteriaceae* can become resistant to carbapenems, there are different approaches to treat infections caused by these bacteria, which include the repurposing of already existing antibiotics, dual therapies with these antibiotics, and the development of new ß-lactamase inhibitors and antibiotics.

## 1. Introduction

Antibiotic resistance occurs when bacteria causing an infection survive after being exposed to a drug that, under normal conditions, would kill it or inhibit its growth [1]. As a result, these surviving strains multiply and spread due to the lack of competition from other strains sensitive to the same drug. Due to the inappropriate prescription and administration of antibiotics, resistant bacteria have become a public health threat worldwide [2]. In fact, the issue of antibiotic resistant bacteria is such that, according to the World Health Organization (WHO) predictions, if antibiotic resistance continues to increase at this rate, infections caused by resistant bacteria will become the top cause of death worldwide, ahead of cancer, diabetes and cardiovascular diseases [3].

In 2017, WHO published a list of antibiotic resistant bacteria against which there is an urgent need to develop new antibiotics [4]. This list is divided into three categories depending on the urgency with which new antibiotics are needed: critical, high and medium priority. Within the critical priority group are carbapenem and 3rd generation cephalosporin resistant *Enterobacteriaceae*. These bacteria are common pathogens causing severe infections such as bloodstream infections, pneumonia, complicated urinary tract infections and complicated intra-abdominal infections. As a result, antibiotic resistance in *Enterobacteriaceae* has significant clinical and socioeconomic consequences [5,6].

Initially, *Enterobacteriaceae* posed a threat to the public health due to their ability to become resistant to antibiotics by producing extended-spectrum ß-lactamases (ESBLs) [7]. To fight this threat, the medical community turned to drugs such as carbapenems as first-line empirical treatments [8]. This new treatment for resistant bacteria had an unexpected result, as it led to a more serious problem, the emergence of carbapenem-resistant *Enterobacteriaceae* (CRE) [9]. In particular, CRE refer to bacteria belonging to the *Enterobacteriaceae* family that have the ability to survive and grow in the presence of clinically relevant concentrations of carbapenems [10]. Specifically, the Centers for Disease Control and Prevention (CDC) defines CRE as enterobacteria non-susceptible to any carbapenem or documented to produce carbapenemases [11]. 

This review analyzes the epidemiology of CRE as well as current and future treatment options against these increasingly resistant bacteria. Furthermore, it provides an extensive review of the different mechanisms by which *Enterobacteriaceae* develop resistance against carbapenems. The presence of these three aspects in one article could be used as a key tool for a better understanding of this emerging problem and as guidance to elaborate plans to manage the CRE crisis and develop new active drugs more efficiently.

## 2. Mechanisms of Drug Resistance

There are three major mechanisms by which *Enterobacteriaceae* become resistant to carbapenems: enzyme production, efflux pumps and porin mutations [12]. Of these, enzyme production is the main resistance mechanism. Gram-negative bacteria generally develop resistances through the production of ß-lactam-hydrolyzing enzymes [13]. Initially, these enzymes inactivated penicillin, however, as different types of antibiotics were introduced in the treatment of infectious diseases, their spectra extended. Thus, cephalosporinases, ESBLs, metallo-ß-lactamases (MBLs) and other carbapenemases appeared [14]. Generally, CRE are divided into two main subgroups: carbapenemase-producing CRE (CP-CRE) and non-carbapenemase-producing CRE (non-CP-CRE) (Figure 1) [15].

### 2.1. Carbapenemase-producing CRE

CP-CRE can produce a large variety of carbapenemases which can be divided in three groups according to the Ambler classification: class A, class B and class D ß-lactamases [16]. There is a fourth class, Ambler class C, however, its clinical relevance remains unknown [17].

Within class A carbapenemases is the clinically relevant *Klebsiella pneumoniae* carbapenemase (KPC) [18]. This is a plasmid encoded enzyme which actively hydrolyzes carbapenems and is partially inhibited by clavulanic acid [19]. Its clinical relevance is due to the fact that it is the most prevalent and most widely spread worldwide [20] *Enterobacteriaceae* producing KPCs have acquired multidrug resistance to ß-lactams, which limits the therapeutic options to treat infections caused by these bacteria [21]. KPC were originally found in *K. pneumoniae* isolates, however, clinical isolates of KPC-producing *Escherichia coli*, *Klebsiella oxytoca, Salmonella enterica, Citrobacter freundii, Enterobacter aerogenes, Enterobacter cloacae, Proteus mirabillis* and *Serratia marcescens* have been identified [22,23,24,25,26] (Table 1). According to a study by Perez et al. [27], a total of 12 *bla_KPC_* gene variants exist globally.

Another major carbapenemase family belonging to class A are MBLs. These enzymes depend on the interaction with zinc ions in the active site of the enzyme [28]. These enzymes are particularly problematic as they have a high potential for horizontal transfer, they lack clinically useful inhibitors, and they have broad hydrolytic properties that affect most ß-lactam antibiotics except for monobactams [29]. However, MBL resistance is usually associated with multidrug-resistance, with MBL-producing isolates often co-expressing ESBLs, which inactivate monobactams [13]. The most common families of MBLs found in *Enterobacteriaceae* were acquired [17]. These families are the New Delhi metallo-ß-lactamase 1 (NDM-1), Imipenem-resistant *Pseudomonas* (IMP)-type carbapenemases and the Verona integron-encoded metallo-ß-lactamases (VIM) [14]. IMP-type carbapenemases were first detected in Japan during the 1990s and have up to 18 varieties [30]. Similarly, VIM was first isolated in Verona, Italy, in 1997 and consists of 14 members [31]. Both MBLs originated in *P. aeruginosa* and were transferred to *Enterobacteriaceae*. In fact, these MBLs share similarities regarding the plasmids they are carried on and their mechanism of action, as both hydrolyze all ß-lactams except for monobactams and are susceptible to all ß-lactam inhibitors [32]. Regarding NDM-1, it is the most recently discovered MBL. It was isolated in India, which is considered the main reservoir of NDM-producing bacteria [33]. Since then, it has spread worldwide, reaching Europe and the United States through tourists [34]. Currently, NDM is predominant in *K. pneumoniae* and *E. coli* [34]. Studies suggest that most plasmids containing *bla_NDM_* also harbor other resistance determinants encoding different ß-lactamases, quinolone resistance and 16S rRNA methylases which confer resistance to aminoglycosides [35].

The third clinically relevant group of carbapenemases are OXA-48-like, which belong to Ambler class D. Six OXA-48-like variants have been identified, OXA-48 being the most widespread [36]. The remaining variants are: OXA-162, OXA-163, OXA-181, OXA-204 and OXA-232. They are all grouped within the OXA-48-like category because they only differ on one to five amino acid substitutions or deletions [36]. These plasmid-mediated enzymes are primarily found in *K. pneumoniae*, *E. coli*, *C. freundii* and *E. cloacae* [37]. A major concern with these carbapenemases is that no existing inhibitors work against them and they have an extraordinary ability to mutate and expand their activity spectrum [38]. These enzymes are highly active against penicillins, have low activity against carbapenems and intermediate activity against broad-spectrum cephalosporins [17].

### 2.2. Non-Carbapenemase-producing CRE 

Besides carbapenemase production, *Enterobacteriaceae* have alternative mechanisms by which they can present carbapenem resistance. These are unspecific mechanisms which can result in multi-drug resistance, such as the production of other ß-lactamases, porin loss and efflux pump overexpression [14]. These mechanisms generally appear paired among themselves or with carbapenemase-production [39]. In fact, while carbapenemases specifically target carbapenems and other ß-lactam antibiotics, efflux pump expression or porin changes are associate with multi-drug resistance [40]. All three alternative mechanisms aim to block the penetration of the antibiotic within the bacterial cell.

Firstly, *Enterobacteriaceae* can produce different types of ß-lactamases, such as AmpC-type ß-lactamases. These enzymes do not degrade carbapenems [41] but they form a bond with the carbapenem molecule, preventing it from accessing its target [42]. Specifically, the plasmid-encoded AmpC CMY-2 is frequently found in *E. coli* and other *Enterobacteriaceae* worldwide, causing resistance to carbapenems [43].

Secondly, resistance-nodulation-division (RND) efflux pumps are a major mechanism of multi-drug resistance in *Enterobacteriaceae* [44]. Among the different efflux systems, the AcrAB-TolC RND system is the most common [44]. This RND efflux pump, along with the CusABC efflux complex, belongs to *E. coli* [45]. Similarly, *Campylobacter jejuni* presents multi-drug resistance through the expression of the CmeABC complex [45]. These resistant genes can be easily transmitted from one microorganism to another through plasmids [46].

Lastly, alterations of porin synthesis also contribute to blocking penetration of carbapenems into the bacterial cell [47]. These alterations have been described in AmpC- and carbapenemase-producing *K. pneumoniae*, which suggests that changes in porin expression play a key role in the ß-lactam resistance displayed by multi-drug resistant bacteria [48]. Studies suggest that strains that have their porins mutated or their expression modulated typically do not have potential for mobilization into community settings but may proliferate locally within hospitals [49].

## 3. Current Resistance Status

Since the detection of the first strain of CRE in the 1980s [50], CRE has rapidly spread worldwide. Epidemiology studies suggest that different carbapenemases predominate in different areas of the world. For that matter, NDM-1 is the main carbapenemase producing resistance in India, Pakistan and Sri Lanka. On the other hand, KPC-producing *Enterobacteriaceae* are endemic in the United States, Colombia, Argentina, Greece and Italy, while OXA-48-like enzyme-producers are endemic in Turkey, Malta, the Middle-East and North Africa [51] (Figure 2). 

As mentioned earlier, the first case of CP-CRE was isolated in Japan and corresponded to an IMP-producing *Serratia marcescens* [50]. This strain caused a plasmid-mediated outbreak in seven Japanese hospitals, followed by a widespread dissemination of *bla*_IMP-1_-harboring *Enterobacteriaceae* throughout Japan. Since then, 52 variants of IMP genes have been identified and have their endemicity limited to Japan and Taiwan [52]. VIM-type MBLs were described shortly after in *P. aeruginosa* strains [53]. By the early 2000s, cases of VIM-producing *Enterobacteriaceae* were already being reported [17]. *K. pneumoniae* and *E. coli* strains producing VIM-type carbapenemases have their endemicity peak in Greece [28]. However, the major threat of MBL-producing *Enterobacteriaceae* appeared with the discovery of an ST14 *K. pneumoniae* strain producing the NDM enzyme from a Swedish patient who received healthcare in New Delhi, India [54]. Bacteria producing this enzyme is endemic in the Indian subcontinent and generally appears as sporadic cases in the rest of the world [55]. NDM-1 producing *Enterobacteriaceae* have been reported both in hospital and community-acquired infections, including urinary tract infections, septicemia, pulmonary infections, peritonitis, device-associated infections and soft tissue infections [56]. An additional issue with NDM-producing bacteria is their ability to spread via environmental sources in community settings of lower-income countries. In fact, studies carried out in India found that 4% of the drinking water and 30% of seepage samples contained NDM-1-producing bacteria [33].

KPC-producing *Enterobacteriaceae* are categorized as one of the most successful pandemics in the history of Gram-negative bacteria, particularly due to *K. pneumoniae* ST258 [57]. This strain has been reported as endemic in Greece, Israel, Latin America and the United States [39]. The endemic state of KPC-producing *Enterobacteriaceae* is not surprising, seeing as the first case of *K. pneumoniae* producing this enzyme was reported in a patient in a North Carolina hospital in 1996 [18]. Only five years later, an outbreak of KPC-producing bacteria took place throughout northeastern United States within hospitalized patients [58]. On the other hand, Greece has one of the highest CRE rates worldwide. Initially, this resistance was due to VIM enzymes, however, in 2007, a rapid dissemination of KPC-producing bacteria made KPC the main mechanism of resistance against carbapenems in the country [39]. Current studies suggest that around 40% of the carbapenemase-resistant *K. pneumoniae* harbor *bla*_KPC_ in Greece [59]. Colombia was the first country within Latin America to report an outbreak of KPC-producing *K. pneumoniae*, which originated from a patient who had travelled to Israel [60]. Since then, Argentina, Chile, Mexico and Brazil have also reported the detection of KPC-producing CRE [39]. 

Finally, regarding OXA-48-like-producing CRE, outbreaks caused by these bacteria have been reported in several countries, however, only Turkey, Japan and Taiwan have reported endemicity [61].

## 4. Treatment Options

Carbapenems continue to be used for the treatment of infections caused by *Enterobacteriaceae* as suggested by both, EUCAST (European Committee on Antimicrobial Susceptibility Testing) and CLSI (Clinical and Laboratory Standards Institute) guidelines [62,63]. The clinical breakpoints of the carbapenems currently used are presented in Table 2. It must be noted that doripenem has been removed from 2019 EUCAST guidelines due to the lack of availability of this drug in most countries. In those countries where doripenem is still available, 2018 EUCAST guidelines must be used as a reference [64]. However, CRE are an increasingly common issue in the clinical practice, rendering carbapenems useless.

Bearing in mind the different mechanisms by which *Enterobacteriaceae* can become resistant to carbapenems, there are different approaches to treat infections caused by these bacteria. These treatment options include the repurposing of already existing antibiotics, dual therapies with these antibiotics and the development of new ß-lactamase inhibitors and antibiotics [65] (Table 3).

Firstly, certain “old antibiotics” which have been included in the therapeutic arsenal for years are still effective against CRE. For example, fosfomycin, frequently used to treat urinary tract infections (UTIs), continues to be effective against approximately 80% of CRE [66]. Similarly, aminoglycosides are still considered first-line therapy for the treatment of carbapenem-resistant *K. pneumoniae* infections [6]. While gentamicin is the most frequently used aminoglycoside, studies report cases where amikacin was the only active molecule [67]. Colistin also remains as a key drug in the treatment of CRE infections [65]. However, CRE, and more particularly *K. pneumoniae*, have started to develop resistance against this drug, decreasing its efficiency as a monotherapy treatment [68]. As a result, colistin has been included as part of a dual therapy with meropenem, which results in a significant reduction of mortality, especially in patients with septic shock, high mortality score or rapidly fatal underlying diseases [69]. Moreover, polymyxins continue to be considered last resort drugs due to their adverse effects, which include nephrotoxicity, neurotoxicity and skin pigmentation [65].

Tigecycline also remains as an option for CRE treatment in certain cases [70]. The particularity with this drug is that it displays low serum concentrations in the approved dosing regimen for the treatment of community-acquired and nosocomial-acquired pneumonia, which hampers clinical outcomes [71]. As a result, a high-dose tigecycline regimen has been investigated and is being used to treat CRE infections. This therapy consists of a 200 mg initial dose and a maintenance dose of 100 mg every 12 h [65]. This high-dose is particularly effective for the treatment of ventilator-associated pneumonia caused by CRE [72]. Furthermore, a systematic review comprising 25 studies reporting the efficacy and safety of tigecycline-based regimens for treating CRE infections concluded that a much lower mortality rate resulted from high-dose tigecycline than standard-dose tigecycline [70]. 

Lastly, carbapenems continue to be used for the treatment of CRE infections. This is done through the combination of two different carbapenems, which is known as “double carbapenems”. Generally, the combination consists of an initial dose of ertapenem followed by a prolonged infusion of meropenem or doripenem over 3 or 4 h with additional 2 g doses of meropenem every 8 h [73]. This therapy is effective against CRE because the greater affinity of ertapenem to KPC makes it play a “sacrificial role”, meaning that it is preferentially hydrolyzed by the carbapenemase, allowing the concomitant administration of the second carbapenem to sustain a high concentration [74]. Comparator studies such as those by Oliva et al. [75] and Venugopalan et al. [76] confirm the efficacy of dual carbapenem therapy, reporting clinical success rates of more than 70% in both cases.

Regarding novel antibacterial drugs, they can be differentiated in two groups: newly approved antibiotics and molecules in development stages. The latest antibiotics approved and already being used to treat CRE infections are ceftazidime/avibactam, meropenem/vaborbactam, plazomicin and eravacycline.

Ceftazidime/avibactam (Allergan) is a novel ß-lactam/ß-lactamase inhibitor combination. The novelty of this combination relies on avibactam, which is a synthetic non-ß-lactam ß-lactamase inhibitor active against ß-lactamases from Ambler classes A, C and D [77]. Clinical studies using this combination are still scarce, however, initial results show an improved mortality rate of 9% compared to the 32% obtained when using colistin [78]. Regardless of the promising initial results, ceftazidime/avibactam resistant strains have already been reported during treatment [79,80]. The resistance is due to mutations in the *bla*_KPC-2_ and *bla*_KPC-3_ genes affecting omega loop D179Y, down-regulation of ompk35/36 and increase in efflux, which could decrease meropenem activity [81]. This should be taken into account by clinicians when prescribing this treatment.

Similarly, meropenem/vaborbactam (Melinta) is also a new ß-lactam/ß-lactamase inhibitor consisting of a carbapenem and a novel boron-containing serine-ß-lactamase inhibitor that potentiates the activity of meropenem [65]. This combination inhibits Ambler classes A and C serine carbapenemases [82]. There are few clinical data with this combination, however, in vivo results showed that, out of 991 clinical isolates of KPC-producing *Enterobacteriaceae*, 99% were susceptible to meropenem-vaborbactam [83]. Furthermore, results from the Tango II trial, which compared the efficacy and safety of this combination with the best available therapy in CRE infections, showed a higher clinical cure (65.6% vs 33.3%) and 28-day mortality (15.6% vs 33.3%) for meropenem/vaborbactam [84].

Plazomicin (Achaogen) is a next-generation semisynthetic aminoglycoside with activity against bacteria producing aminoglycoside-modifying enzymes [85]. Studies report higher potency of plazomicin compared to other aminoglycosides against KPC-producing *Enterobacteriaceae* [86]. Along these lines, Endimiani et al. [86] analyzed collections of clinically relevant KPC-producers with resistance to aminoglycosides and observed inhibition using plazomicin, with a minimum inhibitory concentration (MIC_90_) of ≤2 mg/L [87]. Plazomicin has shown broad-spectrum activity against Gram-positive cocci and Gram-negative bacilli [87], however, MBL-producers are resistant to this antibiotic due to the methyltransferase enzymes which are commonly found, especially in NDM-producers [87]. Aminoglycosides are not generally used as monotherapy, however, the broad spectrum of activity along with the low renal toxicity of plazomicin make it an option for a targeted monotherapy against extensively-drug resistant *Enterobacteriaceae* causing urinary tract infections [88].

Lastly, eravacycline (Tetraphase) is a synthetic fluorocycline with broad-spectrum antimicrobial activity against Gram-positive, Gram-negative and anaerobic bacteria, regardless of resistance to other antibiotic classes [89]. This antibiotic has several potential advantages over tigecycline, which include a more potent in vitro antibacterial activity, excellent oral bioavailability, lower potential for drug interactions and superior activity in biofilm [90]. This drug was also studied in cUTI (complicated urinary tract infection) in two Phase 3 trials (IGNITE 2/3), failing to meet endpoints in both studies, which could be explained by an erratic pharmacokinetic in urine [91]. However, eravacycline did meet endpoints in the IGNITE 4 Phase 3 study, in which it demonstrated similar activity to ertapenem (100% cure rate for eravacycline vs 92.3% for ertapenem) in the treatment of complicated intra-abdominal infections [92]. 

In addition to these already approved drugs, there are six molecules in early developmental stages: imipenem/cilastatin and relebactam (Merck), cediferocol (Shionogi), SPR741 (SperoTherapeutics), zidebactam (Wockhardt), nacubactam (Roche) and VNRX 5133 (VenatoRx Pharmaceuticals). Firstly, imipenem/cilastatin and relebactam shares similarities with previously discussed combinations in that it combines an approved carbapenem with a novel ß-lactamase inhibitor. In fact, the inhibitory mechanism of relebactam is similar to that of avibactam, as it covalently and reversibly binds to classes A and C ß-lactamases [93]. By including relebactam, the activity of imipenem increases considerably against carbapenemase-producing bacteria, up to >16 fold [94]. In fact, the RESTORE-IMI 1 study proved this combination to be as effective and better tolerated than colistin/imipenem for the treatment of infections caused by KPC-producing *Enterobacteriaceae* [95]. Regarding cefiderocol, it is the first siderophore-conjugated cephalosporin antibiotic to advance into late-stage development. This drug has a novel mechanism of action in which the cathecol substituent forms a chelating complex with iron, acting as a trojan horse by using iron active transport systems in gram negative bacteria to bypass the other membrane permeability barrier [96]. This molecule demonstrates potent in vitro and in vivo activity against a variety of Gram-negative bacteria, including CRE [96]. A study analyzed the activity of cefiderocol and comparative agents against 1,022 isolates of carbapenem-nonsusceptible *Enterobacteriaceae*, obtaining MIC_50_ and MIC_90_ for cefiderocol of 1 and 4 µg/mL, respectively [97]. SPR741 is in very early stages of the development process. This molecule is a polymyxin B potentiator that increases ceftazidime and piperazine/tazobactam activity against CRE and ESBLs including OXA-48 [98].

The remaining molecules under development are ß-lactamase inhibitors. Firstly, zidebactam and nacubactam have high affinity to Ambler classes A and C ß-lactamases [99]. Moreover, they also have affinity to PBPs as well as ß-lactam enhancer activity [100]. The cefepime/zidebactam combination is currently in phase 2 clinical trials for the treatment of Gram-negative bacteria. This combination showed potent in vitro activity against carbapenemase-producing *Enterobacteriaceae*, with MIC_50_ of 0.25 mg/L for KPC-producers and 0.5 mg/L for MBL-producers [101]. On the other hand, nacubactam in combination with meropenem is currently in phase 1 trials against Gram-negative bacteria causing UTI infections [102]. Results from this study show improved MIC values for the meropenem/nacubactam combination in comparison with meropenem alone. Furthermore, this combination was active against ceftazidime/avibactam-resistant isolates. Lastly, VNRX 5133 is a cyclic boronate broad spectrum ß-lactamase inhibitor in clinical development with cefepime for the treatment of multidrug-resistant bacteria [103].

## 5. Conclusions

As highlighted by the Global Priority List published by WHO, carbapenem-resistant *Enterobacteriaceae* pose an exponentially increasing threat for the public health worldwide. These bacteria possess diverse and versatile mechanisms of drug resistance, which makes control and early detection of infections caused by CRE difficult. As a result, a joint effort must be made between the scientific and medical community to slow down the appearance of resistances. Along these lines, there is an urgent need to develop new therapeutic guidelines to treat CRE infections. This includes the repurposing of already existing antibiotics such as fosfomycin, aminoglycosides and colistin and the development of novel drugs such as plazomicin, eravacycline or cefiderocol among others.

## Figures and Tables

**Figure 1 antibiotics-08-00122-f001:**
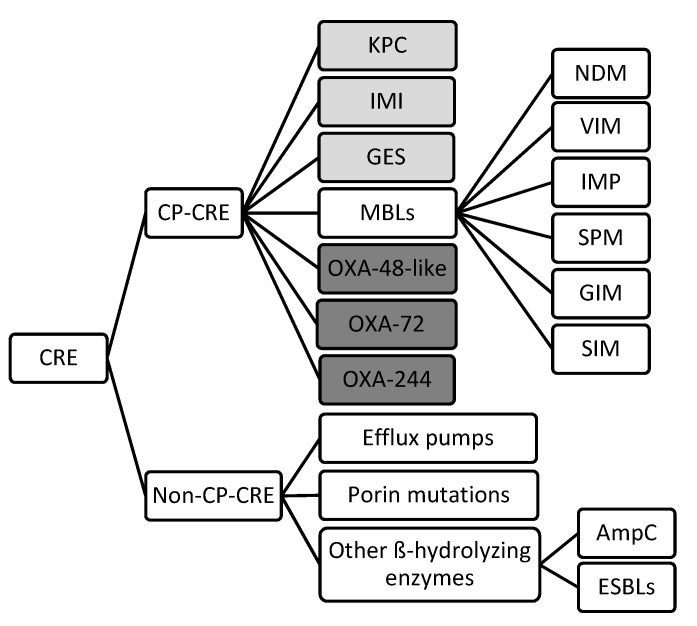
Classification of the different mechanisms of drug resistance in CRE. (Light grey: Ambler class A, White: Ambler class B, Dark grey: Ambler class D) (CRE: Carbapenem-resistant Enterobacteriaceae; CP: carbapenemase producing; KPC: *Klebsiella pneumoniae* carbapenemase; IMI: Imipenem-hydrolyzing ß-lactamase; GES: Guiana extended-spectrum ß-lactamase; MBLs: Metallo-ß-lactamase; OXA: oxacillinase; NDM: New Delhi metallo-ß-lactamase; VIM: Verona integron-borne metallo-ß-lactamase; IMP: Imipenem-resistant *Pseudomonas* carbapenemase; SMP: Sao Paulo metallo-ß-lactamase; GIM: German imipenemase; SIM: Seoul imipenemase; AmpC: Type C ampicillinase; ESBLs: Extended-spectrum ß-lactamase).

**Figure 2 antibiotics-08-00122-f002:**
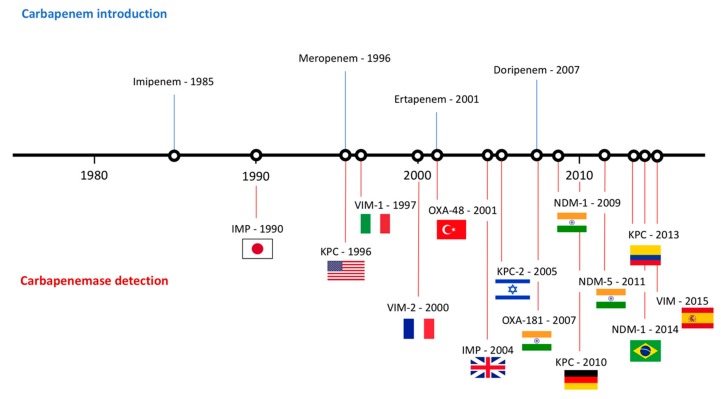
Timeline representing the introduction of carbapenems and the appearance of carbapenemases worldwide.

**Table 1 antibiotics-08-00122-t001:** Carbapenemases detected in different species belonging to the *Enterobacteriaceae* family.

Species	Class A	Class B (MBLs)	Class D	Ref.
*Klebsiella pneumoniae*	KPC-3	NDM-1, VIM-1	OXA-48	Okoche et al. Boutal et al.
*Klebsiella oxytoca*			OXA-48, OXA-181	Okoche et al. Boutal et al.
*Escherichia coli*	KPC	NDM-1, NDM-5, NDM-9, VIM	OXA-48, OXA-181, OXA-244	Okoche et al. Boutal et al.
*Proteus mirabilis*	KPC		OXA-48	Okoche et al. Boutal et al.
*Serratia marcescens*	KPC	VIM		Okoche et al. Boutal et al.
*Enterobacter cloacae*	KPC, IMI-1	VIM-4	OXA-48	Okoche et al. Boutal et al.
*Enterobacter aerogenes*	KPC		OXA-48	Okoche et al. Boutal et al.
*Citrobacter freundii*		VIM	OXA-48	Okoche et al. Boutal et al.
*Citrobacter koseri*			OXA-48	Okoche et al. Boutal et al.
*Salmonella enterica*	KPC-2	NMD-1, NMD-5, VIM-1, VIM-2, IMP-4	OXA-48	Fernández et al.
*Morganella morganii*		NDM-1	OXA-48	Boutal et al.
*Providencia stuartii*	KPC-2	VIM-1		Abdallah et al.
*Providencia rettgeri*		IMP-1	OXA-72	Abdallah et al.

**Table 2 antibiotics-08-00122-t002:** Breakpoints for carbapenems against *Enterobacteriaceae* family.

Antibiotic	Guidelines	Disk Content (µg)	Disk Diffusion (mm)			Dilution (µg/mL)		
			S	I	R	S	I	R
Ertapenem	EUCAST ^1^ CLSI ^2^	10	≥25 ≥23	- 19–21	≤25 ≤18	≤0.5 ≤0.5	- 1	0.5 ≥2
Imipenem	EUCAST ^1^ CLSI ^2^	10	22 ≥23	21–18 20–22	≤17 ≤19	≤2 ≤1	3 2	4 ≥4
Meropenem	EUCAST ^1^ CLSI ^2^	10	22 ≥23	21–17 20–22	16 ≤19	≤2 ≤1	3–7 2	8 ≥4
Doripenem	EUCAST ^3^ CLSI ^2^	10 10	22 ≥23	21–17 20–22	≤16 ≤19	≤1 ≤1	2–3 2	4 ≥4

^1^ The European Committee on Antimicrobial Susceptibility Testing. Breakpoint tables for interpretation of minimum inhibitory concentrations (MICs) and zone diameters. Version 9.0, 2019. Available on: http//www.eucast.org [62]. ^2^ CLSI. Performance Standards for Antimicrobial Susceptibility Testing. 29th ed. CLSI supplement M100. Wayne, PA: Clinical and Laboratory Standards Institute; 2019 [63]. ^3^ The European Committee on Antimicrobial Susceptibility Testing. Breakpoint tables for interpretation of MICs and zone diameters. Version 8.1, 2018. Available on: http//www.eucast.org [64].

**Table 3 antibiotics-08-00122-t003:** Current and future treatment options for infections caused by CRE.

	Drug(Pharmaceutical Company)	Action Mechanism	Structure	Limitations	Ref.
**“Old Antibiotics”**	Fosfomycin(Merck)	Cell wall synthesis inhibitor	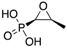	Appearance of resistance	Vardakas et al.
Aminoglycosides	Protein synthesis inhibitor	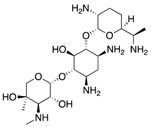	Appearance of resistance	Rodriguez-Bano et al.Satlin et al.
Colistin(Kobayashi Bacteriological Laboratory)	Cell membrane disruptor	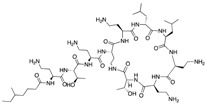	Nephrotoxicity and other severe adverse effects	Karaiskos et al.Daikos et al.
Tigecycline(Pfizer)	Protein synthesis inhibitor	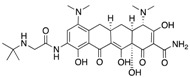	Low concentration in tissue	Ni et al.
**Dual Therapies**	Ertapenem+Meropenem/Doripenem	Cell wall synthesis inhibitor	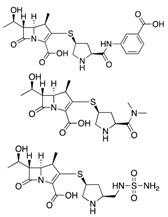	-	Bulik et al.
Ceftazidime/Avibactam(Allergan)	Cell wall synthesis inhibitor/ß-lactamase inhibitor	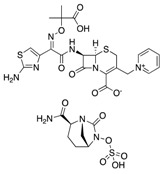	Appearance of resistance	De Jonge et al.
Meropenem/Vaborbactam(Melinta)	Cell wall synthesis inhibitor/ß-lactamase inhibitor	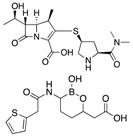	Insufficient clinical data	Karaiskos et al.
**Novel Drugs**	Plazomicin(Achaogen)	Protein synthesis inhibitor	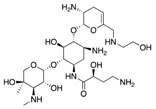	Ineffective against MBL-producers	Landman et al.
Eravacycline(Tetraphase)	Protein synthesis inhibitor	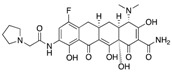	Currently in clinical trials	Zhanel et al.
Imipenem/Relebactam(Merck)	Cell wall synthesis inhibitor/ß-lactamase inhibitor	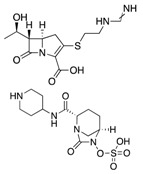	Currently in clinical trials	Blizzard et al.
Cefiderocol(Shionogi)	Cell wall synthesis inhibitor	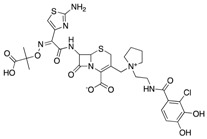	Currently in clinical trials	Saisho et al.
Zidebactam(Wockhardt)	ß-lactamase inhibitor	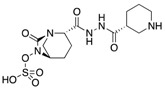	Currently in clinical trials	Karaiskos et al.
Nacubactam(Roche)	ß-lactamase inhibitor	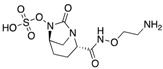	Currently in clinical trials	Papp-Wallace et al.

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
