# Peer review of "Present and Future of Carbapenem-resistant Enterobacteriaceae (CRE) Infections"

_antibiotics, 2019, doi:10.3390/antibiotics8030122_

Round 1

Reviewer 1 Report

The authors have conducted a thorough review of the literature regarding carbapenem-resistant Enterobacteriaceae (CRE) infections including mechanisms of resistance and treatment options. It is a well written paper - thank you for the contribution to this extremely important public health / infectious disease topic.

MAJOR COMMENTS:

Consider adding a timeline for new antimicrobial agents with regards to place in development (i.e. Phase 2 / 3, or if PDUFA date is known – would be good to share with readers). Also consider adding the pharmaceutical companies affiliated with the products (i.e. Ceftaz/Avi (Allergan), meropenem/vaborbactam (Mellinta), eravacycline (Tetraphase), nacubactam (Roche), cefiderocol (Shionogi) with PDUFA expected 4Q2019, etc.) Line 217: Consider expanding on the fosfomycin section: Infecto Pharma at ECCMID 2018, presented data on fosfomycin 3 g PO daily x 7 days being non-inferior to 3 g PO every other day vs. KPCs (fosfomycin in combination with carbapenems or tigecycline). Line 221: Colistin: consider adding comment about polymyxins being last resort due to adverse events (nephrotoxicity >>> neurotoxicity >> skin pigmentation) and other commercially available options (i.e. ceftolozane/tazobactam, ceftazidime/avibactam, and meropenem/vaborbactam) Line 228: Tigecycline: clarify about PK data Re: low serum concentrations in approved dosing regimen (for which indication)? Line 236: refer to “double carbapenems” as this is a quoted strategy in the literature, also consider explaining the mechanism of ertapenem’s “sacrificial role” and the second carbapenem’s strategic role in the attack of the CRE Line 253: Consider expanding on ceftaz/avi resistance mechanisms (i.e. mutations in KPC2/KPC3 affecting omega loop D179Y, down-regulate ompk35/36, and increase in efflux that can decrease meropenem activity – ECCMID 2018, Humphries, S0386) Line 276-282: Consider mentioning how eravacycline was studied in cUTI in two Phase 3 trials (IGNITE 2/3) and its failure to meet endpoints in both studies (could be explained by erratic PK in urine?). IGNITE 4 Phase 3 study of eravacycline vs. meropenem for cIAI did meet endpoints. Line 285: Consider expanding on imi/rel’s activity (more active vs. resistant pseudomonas which differentiates it from meropenem/vabor). In RESTORE-IMI study vs. colistin against imi-nonsusceptible pathogens including pseudomonas and KPC-positive Enterobacteriaceae. Line 289-293: Consider highlighting the unique and novel mechanism of action of the siderophore cephalosporin with a catechol substituent that forms a chelating complex with iron. It acts as a trojan horse and uses iron active transport system in gram negative bacteria to bypass the outer membrane permeability barrier. Also consider discussing the durability of cefiderocol and the potential for resistance (as there is no data to date available about this)? Add to section on line 295 RE: new abx:

o   VNRX 5133 from Venato Rx, is a cyclic boronate broad spectrum B-lactamase inhibitor (similar to vaborbactam) in clinical development with cefepime for ESBL/CRE/MBL and pseudomonas treatment.

o   SPR741 (SperoTherapeutics) – polymyxin B potentiator that increases ceftazidime and pip/tazo activity against CRE/ESBLs including OXA-48.

MINOR COMMENTS:

Syntax comments Line 31: “strains sensible to the same drug” – consider changing to “sensitive” Line 35: “become the first cause of death” – consider changing to “top cause of death” Line 38: “divided in 3 categories” – consider changing to “divided into three categories” Line 82/92: consider removing colloquial language (i.e. “seeing” – ok without this word) Line 83: consider changing to “have broad hydrolytic properties that affect most β-lactam antibiotics…” Line 114/115: “present” is used twice in the same sentence – consider “Enterobacteriaceae have alternatives…” Line 127: Figure 1 – provide abbreviation legend (i.e. GES) Line 139: consider changing to “contribute to blocking penetration of carbapenems…” Line 163/164: consider changing to “Enterobacteriaceae were already being reported…”

Author Response

Response to Reviewer 1 Comments

The authors have conducted a thorough review of the literature regarding carbapenem-resistant Enterobacteriaceae (CRE) infections including mechanisms of resistance and treatment options. It is a well written paper - thank you for the contribution to this extremely important public health / infectious disease topic.

The authors would like to express their gratitude to the reviewer for the time devoted to improve the present amended manuscript.

MAJOR COMMENTS:

Consider adding a timeline for new antimicrobial agents with regards to place in development (i.e. Phase 2 / 3, or if PDUFA date is known – would be good to share with readers). Also consider adding the pharmaceutical companies affiliated with the products (i.e. Ceftaz/Avi (Allergan), meropenem/vaborbactam (Mellinta), eravacycline (Tetraphase), nacubactam (Roche), cefiderocol (Shionogi) with PDUFA expected 4Q2019, etc.)

The suggested data have been included in the text.

Line 217: Consider expanding on the fosfomycin section: Infecto Pharma at ECCMID 2018, presented data on fosfomycin 3 g PO daily x 7 days being non-inferior to 3 g PO every other day vs. KPCs (fosfomycin in combination with carbapenems or tigecycline).

The reference mentioned makes reference to KPC-producing Pseudomonas. The authors consider this information should not be included in the review, as it is focused on Enterobacteriaceae.

Line 221: Colistin: consider adding comment about polymyxins being last resort due to adverse events (nephrotoxicity >>> neurotoxicity >> skin pigmentation) and other commercially available options (i.e. ceftolozane/tazobactam, ceftazidime/avibactam, and meropenem/vaborbactam)

The comment has been included: Moreover, polymyxins continue to be considered last resort drugs due to their adverse effects, which include nephrotoxicity, neurotoxicity and skin pigmentation [61].”

Line 228: Tigecycline: clarify about PK data Re: low serum concentrations in approved dosing regimen (for which indication)?

This sentence has been clarified: The particularity with this drug is that it displays low serum concentrations in the approved dosing regimen for the treatment of community-acquired and nosocomial-acquired pneumonia, which hampers clinical outcomes [67].”

Line 236: refer to “double carbapenems” as this is a quoted strategy in the literature, also consider explaining the mechanism of ertapenem’s “sacrificial role” and the second carbapenem’s strategic role in the attack of the CRE

The mechanism by which this double therapy works has been explained with further detail: Lastly, carbapenems continue to be used for the treatment of CRE infections. This is done through the combination of two different carbapenems, which is known as “double carbapenems”. Generally, the combination consists of an initial dose of ertapenem followed by a prolonged infusion of meropenem or doripenem over 3 or 4 h with additional 2 g doses of meropenem every 8 h [69]. This therapy is effective against CRE because the greater affinity of ertapenem to KPC makes it play a “sacrificial role”, meaning that it is preferentially hydrolyzed by the carbapenemase, allowing the concomitant administration of the second carbapenem to sustain a high concentration [70].”

Line 253: Consider expanding on ceftaz/avi resistance mechanisms (i.e. mutations in KPC2/KPC3 affecting omega loop D179Y, down-regulate ompk35/36, and increase in efflux that can decrease meropenem activity – ECCMID 2018, Humphries, S0386)

The suggested data have been included in the text: The resistance is due to mutations in the blaKPC-2 and blaKPC-3 genes affecting omega loop D179Y, down-regulation of ompk35/36 and increase in efflux, which could decrease meropenem activity [77]. This should be taken into account by clinicians when prescribing this treatment.”

Line 276-282: Consider mentioning how eravacycline was studied in cUTI in two Phase 3 trials (IGNITE 2/3) and its failure to meet endpoints in both studies (could be explained by erratic PK in urine?). IGNITE 4 Phase 3 study of eravacycline vs. meropenem for cIAI did meet endpoints.

The suggested data have been included in the text: This drug was also studied in cUTI (complicated urinary tract infection) in two Phase 3 trials (IGNITE 2/3), failing to meet endpoints in both studies, which could be explained by an erratic pharmacokinetic in urine [87]. However, eravacycline did meet endpoints in the IGNITE 4 Phase 3 study, in which it demonstrated similar activity to ertapenem (100% cure rate for eravacycline vs 92.3% for ertapenem) in the treatment of complicated intra-abdominal infections [88].”

Line 285: Consider expanding on imi/rel’s activity (more active vs. resistant pseudomonas which differentiates it from meropenem/vabor). In RESTORE-IMI study vs. colistin against imi-nonsusceptible pathogens including pseudomonas and KPC-positive Enterobacteriaceae. 

The suggested data have been included in the text: In fact, the RESTORE-IMI 1 study proved this combination to be as effective and better tolerated than colistin/imipenem for the treatment of infections caused by KPC-producing Enterobacteriaceae [91].“

Line 289-293: Consider highlighting the unique and novel mechanism of action of the siderophore cephalosporin with a catechol substituent that forms a chelating complex with iron. It acts as a trojan horse and uses iron active transport system in gram negative bacteria to bypass the outer membrane permeability barrier. Also consider discussing the durability of cefiderocol and the potential for resistance (as there is no data to date available about this)?

The suggested data have been included in the text: Regarding cefiderocol, it is the first siderophore-conjugated cephalosporin antibiotic to advance into late-stage development. This drug has a novel mechanism of action in which the cathecol substituent forms a chelating complex with iron, acting as a trojan horse by using iron active transport systems in gram negative bacteria to bypass the other membrane permeability barrier [92].”

Add to section on line 295 RE: new abx:

o   VNRX 5133 from Venato Rx, is a cyclic boronate broad spectrum B-lactamase inhibitor (similar to vaborbactam) in clinical development with cefepime for ESBL/CRE/MBL and pseudomonas treatment.

o   SPR741 (SperoTherapeutics) – polymyxin B potentiator that increases ceftazidime and pip/tazo activity against CRE/ESBLs including OXA-48.

These new antibiotics have been added: SPR741 is in very early stages of the development process. This molecule is a polymyxin B potentiator that increases ceftazidime and piperazine/tazobactam activity against CRE and ESBLs including OXA-48 [94].”

“Lastly, VNRX 5133 is a cyclic boronate broad spectrum ß-lactamase inhibitor in clinical development with cefepime for the treatment of multidrug-resistant bacteria [99].”

MINOR COMMENTS:

Syntax comments

Line 31: “strains sensible to the same drug” – consider changing to “sensitive”

It has been changed.

Line 35: “become the first cause of death” – consider changing to “top cause of death”

It has been changed.

Line 38: “divided in 3 categories” – consider changing to “divided into three categories”

It has been changed.

Line 82/92: consider removing colloquial language (i.e. “seeing” – ok without this word)

It has been changed.

Line 83: consider changing to “have broad hydrolytic properties that affect most β-lactam antibiotics…”

It has been changed.

Line 114/115: “present” is used twice in the same sentence – consider “Enterobacteriaceae have alternatives…”

It has been changed.

Line 127: Figure 1 – provide abbreviation legend (i.e. GES)

All abbreviation legends have been added.

Line 139: consider changing to “contribute to blocking penetration of carbapenems…”

It has been changed.

Line 163/164: consider changing to “Enterobacteriaceae were already being reported…” 

It has been changed.

Reviewer 2 Report

In the present review, Suay-García et al. discussed the epidemiology of carbapenem-resistant Enterobacteriaceae (CRE) as well as current and future treatment options against these increasingly resistant bacteria. The review mainly focused on the important mechanisms by which Enterobacteriaceae become resistant to carbapenem and the current approaches to treat the infection. This review has been well written. However, there are some sections that need to be discussed before it can be considered for publication.

Comments:

There are several reviews regarding management or control of carbapenem-resistant Enterobacteriaceae. Authors should highlight more on the points that could distinguish this review from previous reviews in the abstract or introduction. Authors should discuss the currently available detection method for CP-CRE and non-CP-CRE bacteria. Authors should provide a schematic diagram to depict the major mechanisms that allow Enterobacteriaceae to resist carbapenem. The authors covered extensively the major agents that are either in use or currently under development to treat the disease. Yet, there seems to be limitations associated with these agents, which hamper their clinical use. Authors should consider summarizing such limitations/disadvantages of those agents in a table. This will help the reader to keep track of the pros and cons of each treatment option and allow better appreciation on the need to develop newer therapeutics. Please provide full form of MIC.

Author Response

Response to Reviewer 2 Comments

Comments and Suggestions for Authors

In the present review, Suay-García et al. discussed the epidemiology of carbapenem-resistant Enterobacteriaceae (CRE) as well as current and future treatment options against these increasingly resistant bacteria. The review mainly focused on the important mechanisms by which Enterobacteriaceae become resistant to carbapenem and the current approaches to treat the infection. This review has been well written. However, there are some sections that need to be discussed before it can be considered for publication.

The authors would like to express their gratitude to the reviewer for the time devoted to improve the present amended manuscript.

Comments:

There are several reviews regarding management or control of carbapenem-resistant Enterobacteriaceae. Authors should highlight more on the points that could distinguish this review from previous reviews in the abstract or introduction.

The authors have included points defending the relevance of this review in the introduction:  Furthermore, it provides an extensive review on the different mechanisms by which Enterobacteriaceae develop resistance against carbapenems. The presence of these three aspects in one article could be used as a key tool for a better understanding of this emerging problem and as guidance to elaborate plans to manage the CRE crisis and develop new active drugs more efficiently.”

Authors should discuss the currently available detection method for CP-CRE and non-CP-CRE bacteria.

We are grateful for the reviewer’s suggestion, however, this review is focused on current resistance status and treatment perspectives. Including detection methods would belong to a more clinical review.

Authors should provide a schematic diagram to depict the major mechanisms that allow Enterobacteriaceae to resist carbapenem.

We consider that the major resistance mechanisms are depicted in figure 1 with sufficient detail.

The authors covered extensively the major agents that are either in use or currently under development to treat the disease. Yet, there seems to be limitations associated with these agents, which hamper their clinical use. Authors should consider summarizing such limitations/disadvantages of those agents in a table. This will help the reader to keep track of the pros and cons of each treatment option and allow better appreciation on the need to develop newer therapeutics.

Limitations of the different treatment options have been included in Table 3.

Please provide full form of MIC. 

It has been added.

Reviewer 3 Report

The review article by Beatriz Suay-García and María Teresa Pérez-Gracia covers the issue of carbapenem-resistant. The authors have done a great job of discussing all aspects of resistance development and available treatments. Authors also emphasize that there is a need for proper guidelines to combat the carbapenem-resistant 2 Enterobacteriaceae (CRE) infections. I only have some minor comments

global threat worldwide> global and worldwide have the same meaning Since then, it has spread worldwide, reaching Europe and the United States through tourists.>>Authors are not citing the appropriate reference to back this statement. The facts based on assumptions should be there in scientific reviews. Tourists may or may not be the reason for disease spread. 200mg> give a space between 200 and mg. This error in multiple places. Hours> Use standard notation. convert to h

Author Response

Response to Reviewer 3 Comments

Comments and Suggestions for Authors

The review article by Beatriz Suay-García and María Teresa Pérez-Gracia covers the issue of carbapenem-resistant. The authors have done a great job of discussing all aspects of resistance development and available treatments. Authors also emphasize that there is a need for proper guidelines to combat the carbapenem-resistant 2 Enterobacteriaceae (CRE) infections.

The authors would like to express their gratitude to the reviewer for the time devoted to improve the present amended manuscript.

I only have some minor comments:

global threat worldwide> global and worldwide have the same meaning

This sentence has been rephrased:  Carbapenem-resistant Enterobacteriaceae (CRE) have become a public health threat worldwide.”

Since then, it has spread worldwide, reaching Europe and the United States through tourists.>>Authors are not citing the appropriate reference to back this statement. The facts based on assumptions should be there in scientific reviews. Tourists may or may not be the reason for disease spread.

The reference has been included: Since then, it has spread worldwide, reaching Europe and the United States through tourists [30].”

200mg> give a space between 200 and mg. This error in multiple places.

The mistakes have been corrected.

Hours> Use standard notation. convert to h 

It has been corrected.
